# A gradient estimator via L1-randomization for online zero-order optimization with two point feedback

**Arya Akhavan**
Istituto Italiano di Tecnologia
CREST, ENSAE, IP Paris
aria.akhavanfoomani@iit.it

**Evgenii Chzhen**
Université Paris-Saclay, CNRS
Laboratoire de mathématiques d'Orsay
evgenii.chzhen@cnrs.fr

**Massimiliano Pontil**
Istituto Italiano di Tecnologia
University College London
massimiliano.pontil@iit.it

**Alexandre B. Tsybakov**
CREST, ENSAE, IP Paris
alexandre.tsybakov@ensae.fr

## Abstract

This work studies online zero-order optimization of convex and Lipschitz functions. We present a novel gradient estimator based on two function evaluations and randomization on the $\ell_1$-sphere. Considering different geometries of feasible sets and Lipschitz assumptions we analyse online dual averaging algorithm with our estimator in place of the usual gradient. We consider two types of assumptions on the noise of the zero-order oracle: canceling noise and adversarial noise. We provide an anytime and completely data-driven algorithm, which is adaptive to all parameters of the problem. In the case of canceling noise that was previously studied in the literature, our guarantees are either comparable or better than state-of-the-art bounds obtained by Duchi et al. [14] and Shamir [33] for non-adaptive algorithms. Our analysis is based on deriving a new weighted Poincaré type inequality for the uniform measure on the $\ell_1$-sphere with explicit constants, which may be of independent interest.

## 1 Introduction

In this work we study the problem of convex online zero-order optimization with two-point feedback, in which adversary fixes a sequence $f_1, f_2, \ldots : \mathbb{R}^d \to \mathbb{R}$ of convex functions and the goal of the learner is to minimize the cumulative regret with respect to the best action in a prescribed convex set $\Theta \subseteq \mathbb{R}^d$. This problem has received significant attention in the context of continuous bandits and online optimization [see e.g., 1, 3, 12, 13, 16, 18, 21, 25, 29, 33, and references therein].

We consider the following protocol: at each round $t = 1, 2, \ldots$ the algorithm chooses $\boldsymbol{x}_t', \boldsymbol{x}_t'' \in \mathbb{R}^d$ (that can be queried outside of $\Theta$) and the adversary reveals

$$f_t(\boldsymbol{x}_t') + \xi_t' \qquad \text{and} \qquad f_t(\boldsymbol{x}_t'') + \xi_t'' \ ,$$

where $\xi_t', \xi_t'' \in \mathbb{R}$ are the noise variables (random or not) to be specified. Based on the above information and the previous rounds, the learner outputs $\boldsymbol{x}_t \in \Theta$ and suffers loss $f_t(\boldsymbol{x}_t)$. The goal of the learner is to minimize the cumulative regret

$$\sum_{t=1}^{T} f_t(\boldsymbol{x}_t) - \min_{\boldsymbol{x} \in \Theta} \sum_{t=1}^{T} f_t(\boldsymbol{x}) \ .$$

At the core of our approach is a novel zero-order gradient estimator based on two function evaluations outlined in Algorithm 1. A key novelty of our estimator is that it employs a randomization step over

36th Conference on Neural Information Processing Systems (NeurIPS 2022).

---

**Algorithm 1:** Zero-Order $\ell_1$-Randomized Online Dual Averaging

---

**Input:** Convex function $V(\cdot)$, step size $\eta_1 > 0$, and parameters $h_t$, for $t = 1, 2, \ldots,$
**Initialization:** Generate independently vectors $\boldsymbol{\zeta}_1, \boldsymbol{\zeta}_2, \ldots$ uniformly distributed on $\partial B_1^d$, and set
$\quad\quad\quad \boldsymbol{z}_1 = \boldsymbol{0}$
**for** $t = 1, \ldots,$ **do**
$\quad \boldsymbol{x}_t = \arg\max_{\boldsymbol{x} \in \Theta} \{\eta_t \langle \boldsymbol{z}_t, \boldsymbol{x} \rangle - V(\boldsymbol{x})\}$
$\quad y_t' = f_t(\boldsymbol{x}_t + h_t \boldsymbol{\zeta}_t) + \xi_t' \quad$ **and** $\quad y_t'' = f_t(\boldsymbol{x}_t - h_t \boldsymbol{\zeta}_t) + \xi_t'' \quad\quad\quad$ // Query
$\quad \boldsymbol{g}_t = \frac{d}{2h_t}(y_t' - y_t'')\,\mathrm{sign}(\boldsymbol{\zeta}_t) \quad\quad\quad\quad\quad$ // $\ell_1$-gradient estimate
$\quad \boldsymbol{z}_{t+1} = \boldsymbol{z}_t - \boldsymbol{g}_t \quad\quad\quad\quad\quad\quad\quad\quad\quad$ // Update $\boldsymbol{z}_t$
$\quad$ update the step-size $\eta_{t+1}$
**end**

---

the $\ell_1$ sphere. This is in contrast to most of the prior work [see e.g., 1, 3–5, 14, 16, 17, 22, 28, 32] that was employing $\ell_2$ or $\ell_\infty$ type randomizations to define $\boldsymbol{x}_t', \boldsymbol{x}_t''$. We use the proposed estimator within an online dual averaging procedure to tackle the zero-order online convex optimization problem, matching or improving the state-of-the-art results. Duchi et al. [14] and Shamir [33] have studied instances of the above problem under the assumption that $\xi_t' = \xi_t''$, which we will further refer to as canceling noise assumption. Specifically, [14] considered the stochastic optimization framework where $f_t = f$, for every $t$, and obtained bounds on the optimization error rather than on cumulative regret, while [33] analyzed the case $\xi_t' = \xi_t'' = 0$. The results in [14, 33] are obtained for the objective functions that are Lipschitz with respect to the $\ell_q$-norm for $q = 1$ and $q = 2$, although, with extra derivations it is possible to extend the above mentioned results beyond such cases. The proposed method allows us to improve upon these results in several aspects.

**Contributions.** The contributions of the present paper can be summarized as follows. **1)** We present a new randomized zero-order gradient estimator and study its statistical properties, both under canceling noise and under adversarial noise (see Lemma 1 and Lemma 4); **2)** In the canceling noise case ($\xi_t' = \xi_t''$) in Theorem 1 we show that dual averaging based on our gradient estimator either improves or matches the state-of-the-art bounds [14, 33]. We derive the results for Lipschitz functions with respect to all $\ell_q$-norms, $q \in [1, \infty]$. In particular, when $q = 1$ and $\Theta$ is the probability simplex, our bound is better by a $\sqrt{\log(d)}$ factor than that of [14, 33]; **3)** We propose a completely data-driven and anytime version of the algorithm, which is adaptive to all parameters of the problem. We show that it achieves analogous performance as the non-adaptive algorithm in the case of canceling noise and only slightly worse performance under adversarial noise. To the best of our knowledge, no adaptive algorithms were developed for zero-order online problems in our setting so far; **4)** As a key element of our analysis, we derive in Lemma 3 a weighted Poincaré type inequality [following the terminology of 10] with explicit constants for the uniform measure on $\ell_1$-sphere. This result may be of independent interest.

**Notation.** Throughout the paper we use the following notation. We denote by $\|\cdot\|_p$ the $\ell_p$-norm in $\mathbb{R}^d$. For any $\boldsymbol{x} \in \mathbb{R}^d$ we denote by $\boldsymbol{x} \mapsto \mathrm{sign}(\boldsymbol{x})$ the component-wise sign function (defined at 0 as 1). We let $\langle \cdot, \cdot \rangle$ be the standard inner product in $\mathbb{R}^d$. For $p \in [1, \infty]$ we introduce the open $\ell_p$-ball and $\ell_p$-sphere respectively as

$$B_p^d \triangleq \left\{ \boldsymbol{x} \in \mathbb{R}^d \,:\, \|\boldsymbol{x}\|_p < 1 \right\} \quad\quad \text{and} \quad\quad \partial B_p^d \triangleq \left\{ \boldsymbol{x} \in \mathbb{R}^d \,:\, \|\boldsymbol{x}\|_p = 1 \right\} \ .$$

For two $a, b \in \mathbb{R}$, we denote by $a \wedge b$ (*resp.* $a \vee b$) the minimum (*resp.* the maximum) between $a$ and $b$. We denote by $\Gamma : (0, \infty) \to \mathbb{R}$, the gamma function. In what follows, $\log$ always stands for the natural logarithm and $e$ is Euler's number.

## 2 The algorithm

Let $\Theta$ be a closed convex subset of $\mathbb{R}^d$ and let $V : \Theta \to \mathbb{R}$ be a convex function. The procedure that we propose in this paper is summarized in Algorithm 1.

**Intuition behind the gradient estimate.** The form of gradient estimator $\boldsymbol{g}_t$ in Algorithm 1 is explained by Stokes' theorem (see Theorem 5 in the appendix and the discussion that follows).

Stokes' theorem provides a connection between the gradient of a function $f$ (first order information) and $f$ itself (zero order information). Under some regularity conditions, it establishes that

$$\int_D \nabla f(\boldsymbol{x})\,\mathrm{d}\boldsymbol{x} = \int_{\partial D} f(\boldsymbol{x})\boldsymbol{n}(\boldsymbol{x})\,\mathrm{d}S(\boldsymbol{x})\ ,$$

where $\partial D$ is the boundary of $D$, $\boldsymbol{n}$ is the outward normal vector to $\partial D$, and $\mathrm{d}S(\boldsymbol{x})$ denotes the surface measure. Introducing $\boldsymbol{U}^D$ and $\boldsymbol{\zeta}^{\partial D}$ distributed uniformly on $D$ and $\partial D$ respectively, we can rewrite the above identity as

$$\mathbf{E}[\nabla f(\boldsymbol{U}^D)] = \frac{\mathrm{Vol}_{d-1}(\partial D)}{\mathrm{Vol}_d(D)} \cdot \mathbf{E}[f(\boldsymbol{\zeta}^{\partial D})\boldsymbol{n}(\boldsymbol{\zeta}^{\partial D})]\ ,$$

where $\mathrm{Vol}_{d-1}(\partial D)$ is the surface area of $D$ and $\mathrm{Vol}_d(D)$ is its volume. In what follows we consider the special case $D = B_1^d$. For this choice of $D$ we have $\boldsymbol{n}(\boldsymbol{x}) = \frac{1}{\sqrt{d}} \cdot \mathrm{sign}(\boldsymbol{x})$ with $\mathrm{Vol}_{d-1}(\partial D)/\mathrm{Vol}_d(D) = d^{3/2}$ leading to our gradient estimate for the two-point feedback setup.

**Computational aspects.** Let us highlight two appealing practical features of the $\ell_1$-randomized gradient estimator $\boldsymbol{g}_t$ in Algorithm 1. First, we can easily evaluate any $\ell_p$-norm of $\boldsymbol{g}_t$. Indeed, it holds that $\|\boldsymbol{g}_t\|_p = (d^{1+1/p}/2h_t)|y_t' - y_t''|$, i.e., computing $\|\boldsymbol{g}_t\|_p$ only requires $O(1)$ elementary operations. Second, this gradient estimator is very economic in terms of the required memory: in order to store $\boldsymbol{g}_t$ we only need $d$ bits and 1 float. None of these properties is inherent to the popular alternatives based on the randomization over the $\ell_2$-sphere [see e.g., 5, 16, 22] or on Gaussian randomization [see e.g., 19, 23, 24].

To compute $\boldsymbol{g}_t$ one needs to generate $\boldsymbol{\zeta}_t$ distributed uniformly on $\partial B_1^d$. The most straightforward way to do it consists in first generating a $d$-dimensional vector of i.i.d. centered scaled Laplace random variables and then normalizing this vector by its $\ell_1$-norm. The result is guaranteed to follow the uniform distribution on $\partial B_d^1$ [see e.g., 30, Lemma 1]. Furthermore, to sample from the centered scaled Laplace distribution one can simply use inverse transform sampling. Indeed, if $U$ is distributed uniformly on $(0,1)$, then $\log(2U)\mathbf{1}(U > 1/2) - \log(2 - 2U)\mathbf{1}(U \geq 1/2)$ follows centered scaled Laplace distribution.

## 3 Assumptions

We say that the convex function $V(\cdot)$ is 1-strongly convex with respect to the $\ell_p$-norm on $\Theta$ if

$$V(\boldsymbol{x}') \geq V(\boldsymbol{x}) + \langle \boldsymbol{w},\, \boldsymbol{x}' - \boldsymbol{x} \rangle + \frac{1}{2}\|\boldsymbol{x} - \boldsymbol{x}'\|_p^2\ ,$$

for all $\boldsymbol{x}, \boldsymbol{x}' \in \Theta$ and all $\boldsymbol{w} \in \partial V(\boldsymbol{x})$, where $\partial V(\boldsymbol{x})$ is the subdifferential of $V$ at point $\boldsymbol{x}$.

Throughout the paper, we assume that $p, q \in [1, \infty]$, $d \geq 3$, and set $p^*, q^* \in [1, \infty]$ such that $\frac{1}{p} + \frac{1}{p^*} = 1$ and $\frac{1}{q} + \frac{1}{q^*} = 1$, with the usual convention $1/\infty = 0$. We will use the following assumptions.

**Assumption 1.** *The following conditions hold:*

1. *The set $\Theta \subset \mathbb{R}^d$ is compact and convex.*

2. *There exists $V : \Theta \to \mathbb{R}$, which is lower semi-continuous, 1-strongly convex on $\Theta$ w.r.t. the $\ell_p$-norm and such that*

$$\sup_{\boldsymbol{x}\in\Theta} V(\boldsymbol{x}) - \inf_{\boldsymbol{x}\in\Theta} V(\boldsymbol{x}) \leq R^2$$

   *for some constant $R > 0$.*

3. *Each function $f_t : \mathbb{R}^d \to \mathbb{R}$ is convex on $\mathbb{R}^d$ for all $t \geq 1$.*

4. *For all $\boldsymbol{x}, \boldsymbol{x}' \in \mathbb{R}^d$, and all $t \geq 1$ we have $|f_t(\boldsymbol{x}) - f_t(\boldsymbol{x}')| \leq L\|\boldsymbol{x} - \boldsymbol{x}'\|_q$ for some constant $L > 0$.*

Assumption 1 is rather standard in the study of dual averaging-type algorithms and have been previously considered in the context of zero-order problems in [14, 33]. We assume that $\Theta$ is compact as we are interested in the worst-case regret, which ensures that $R < +\infty$. We discuss extensions of our results to the case of unbounded $\Theta$ in Section 8. Note that the constant $R > 0$ is not necessarily dimension independent. Below we provide two classical examples of $V$ [see e.g., 31, Section 2].

**Example 1.** *Let $\Theta$ be any convex subset of $\mathbb{R}^d$ and $p \in (1,2]$. Then, $V(\boldsymbol{x}) = \frac{1}{2(p-1)}\|\boldsymbol{x}\|_p^2$ is 1-strongly convex on $\Theta$ w.r.t. the $\ell_p$-norm.*

**Example 2.** *Let $\Theta = \left\{\boldsymbol{x} \in \mathbb{R}^d : \|\boldsymbol{x}\|_1 = 1, \boldsymbol{x} \geq 0\right\}$. Then[1], $V(\boldsymbol{x}) = \sum_{j=1}^d x_j \log(x_j)$ is 1-strongly convex on $\Theta$ w.r.t. the $\ell_1$-norm and $R^2 \leq \log(d)$.*

**Assumptions on the noise.** We consider two different assumptions on the noises $\xi_t', \xi_t''$. The first noise assumption is common in the stochastic optimization context [see e.g., 14, 19, 23, 24, 33].

**Assumption 2** (Canceling noise). *For all $t = 1, 2, \ldots$, it holds that $\xi_t' = \xi_t''$ almost surely.*

Formally, Assumption 2 permits noisy evaluations of function values. However, due to the fact that we are allowed to query $f_t$ at two points, taking difference of $y_t'$ and $y_t''$ in the estimator of the gradient effectively erases the noise. It results in a smaller variance of our gradient estimator. Importantly, Assumption 2 covers the case of no noise, that is, the classical online optimization setting as defined, e.g., in [31].

Second, we consider an adversarial noise assumption, which is essentially equivalent to the assumptions used in [3, 4].

**Assumption 3** (Adversarial noise). *For all $t = 1, 2, \ldots$, it holds that: (i) $\mathbf{E}[(\xi_t')^2] \leq \sigma^2$ and $\mathbf{E}[(\xi_t'')^2] \leq \sigma^2$; (ii) $(\xi_t')_{t\geq 1}$ and $(\xi_t'')_{t\geq 1}$ are independent of $(\boldsymbol{\zeta}_t)_{t\geq 1}$*

Assumption 3 allows for stochastic $\xi_t'$ and $\xi_t''$ that are not necessarily zero-mean or independent over the trajectory. Furthermore, it permits bounded non-stochastic adversarial noises. Part (ii) of Assumption 3 is always satisfied. Indeed, $\xi_t'$'s and $\xi_t''$'s are coming from the environment and are unknown to the learner while $\boldsymbol{\zeta}_t$'s are artificially generated by the learner. We mention part (ii) only for formal mathematical rigor.

Note that, since the choice of function $V$ belongs to the learner and $\Theta$ is given, it is always reasonable to assume that parameter $R$ is known. At the same time, parameters $L$ and $\sigma$ may be either known or unknown. We will study both cases in the next sections.

## 4 Upper bounds on the regret

In this section, we present the main convergence results for Algorithm 1 when $L, \sigma, T$ are known to the learner. The case when they are unknown is analyzed in Section 5, where we develop fully adaptive versions of Algorithm 1.

To state our results in a unified way, we introduce the following sequence that depends on the dimension $d$ and on the norm index $q \geq 1$:

$$\mathrm{b}_q(d) \triangleq \frac{1}{d+1} \cdot \begin{cases} q d^{\frac{1}{q}} & \text{if } q \in [1, \log(d)), \\ e \log(d) & \text{if } q \geq \log(d). \end{cases}$$

The value $\mathrm{b}_q(d)$ will explicitly influence the choice of the step size $\eta > 0$ and of the discretization parameter $h > 0$.

The first result of this section establishes the convergence guarantees under the canceling noise assumption. This case was previously considered by Duchi et al. [14] and Shamir [33].

**Theorem 1.** *Let Assumptions 1 and 2 be satisfied. Then, Algorithm 1 with the parameters*

$$\eta = \frac{\mathtt{A}R}{L}\sqrt{\frac{d^{-1-\frac{2}{q\wedge 2}+\frac{2}{p}}}{T}} \quad \text{and any} \quad h \leq \frac{7R}{100\mathrm{b}_q(d)\sqrt{T}}d^{\frac{1}{2}+\frac{1}{q\wedge 2}-\frac{1}{p}} \;,$$

*where $\mathtt{A} = (\sqrt{6} + \sqrt{12})^{-1}$, satisfies, for any $\boldsymbol{x} \in \Theta$,*

$$\mathbf{E}\left[\sum_{t=1}^T \left(f_t(\boldsymbol{x}_t) - f_t(\boldsymbol{x})\right)\right] \leq 11.9 \cdot RL\sqrt{Td^{1+\frac{2}{q\wedge 2}-\frac{2}{p}}} \;.$$

---

[1]We use the convention that $0\log(0) = 0$.

Note that, as in other related works [14, 20, 23, 24, 33], under the canceling noise (or no noise) assumption the discretization parameter $h > 0$ can be chosen arbitrary small. This is due to the fact that, under the canceling noise assumption, the variance of the gradient estimate $\boldsymbol{g}_t$ is bounded by a constant independent of $h$. It is no longer the case under the adversarial noise assumption as exhibited in the next theorem.

**Theorem 2.** *Let Assumptions 1 and 3 be satisfied. Then Algorithm 1 with the parameters*

$$\eta = \frac{R}{\sqrt{TL}} \left( \frac{\sigma \mathbf{b}_q(d)}{\sqrt{2}R} \sqrt{Td^{4-\frac{2}{p}}} + ALd^{1+\frac{2}{q\wedge 2}-\frac{2}{p}} \right)^{-\frac{1}{2}} \quad and \quad h = \left( \frac{\sqrt{2}R\sigma}{L\mathbf{b}_q(d)} \right)^{\frac{1}{2}} T^{-\frac{1}{4}}d^{1-\frac{1}{2p}} \ ,$$

*where $A = 6(1+\sqrt{2})^2$, satisfies, for any $\boldsymbol{x} \in \Theta$,*

$$\mathbf{E}\left[ \sum_{t=1}^T \big( f_t(\boldsymbol{x}_t) - f_t(\boldsymbol{x}) \big) \right] \leq 11.9 \cdot RL\sqrt{Td^{1+\frac{2}{q\wedge 2}-\frac{2}{p}}}$$

$$+ 2.4 \cdot \sqrt{RL\sigma}T^{\frac{3}{4}} \cdot \begin{cases} \sqrt{qd^{1+\frac{1}{q}-\frac{1}{p}}} & \text{if } q \in [1, \log(d)), \\ \sqrt{e\log(d)d^{1-\frac{1}{p}}} & \text{if } q \geq \log(d). \end{cases}$$

**Comparison to state-of-the-art bounds.** We provide two examples of $p, q, \Theta$ and compare results for our new method to those of [14, 33] where only the canceling noise Assumption 2 and $q \in \{1, 2\}$ were considered.

**Corollary 1.** *Let $p = q = 2$ and $\Theta = B_2^d$. Then under Assumption 2, Algorithm 1 with $V : \Theta \to \mathbb{R}$ defined in Example 1, satisfies*

$$\mathbf{E}\left[ \sum_{t=1}^T \big( f_t(\boldsymbol{x}_t) - f_t(\boldsymbol{x}) \big) \right] \leq 11.9 \cdot L\sqrt{dT} \ .$$

In the setup of Corollary 1, Duchi et al. [14] obtain $O(L\sqrt{dT\log(d)})$ rate and Shamir [33] exhibits $O(L\sqrt{dT})$, which is the optimal rate. Both results do not specify the leading absolute constants.

**Corollary 2.** *Let $p = q = 1$ and $\Theta = \big\{ \boldsymbol{x} \in \mathbb{R}^d \ : \ \boldsymbol{x} \geq 0, \ \|\boldsymbol{x}\|_1 = 1 \big\}$. Then under Assumption 2, Algorithm 1 with $V : \Theta \to \mathbb{R}$ defined in Example 2, satisfies*

$$\mathbf{E}\left[ \sum_{t=1}^T \big( f_t(\boldsymbol{x}_t) - f_t(\boldsymbol{x}) \big) \right] \leq 11.9 \cdot L\sqrt{dT\log(d)} \ .$$

In the setup of Corollary 2, Shamir [33] proves the rate $O(L\sqrt{dT}\log(d))$ for the method with $\ell_2$-randomization. On the other hand, Duchi et al. [14] derived a lower bound $\Omega(\sqrt{dT/\log(d)})$. Thus, our algorithm further reduces the gap between the upper and the lower bounds.

Finally, note that in the case $p = 1, q = 2$ with $V : \Theta \to \mathbb{R}$ defined in Example 2 the bound of Theorem 1 is of the order $O(\sqrt{T\log(d)})$. This case was handled by an algorithm with $\ell_1$-randomization slightly different from ours in [18] leading to the suboptimal rate $O(\sqrt{dT\log(d)})$.

## 5 Adaptive algorithms

Theorems 1 and 2 used the step size $\eta$ and the discretization parameter $h$ that depend on the potentially unknown quantities $L, \sigma$, and the optimization horizon $T$. In this section, we show that, under the canceling noise Assumption 2, adaptation to unknown $L$ comes with nearly no price. On the other hand, under the adversarial noise Assumption 3, our adaptive rate has a slightly worse dependence on $L$ and $\sigma$ in the dominant term. The proof is based on combining the adaptive scheme for online dual averaging [see Section 7.13 in 26, for an overview] with our bias and variance evaluations, cf. Section 6 below.

**Theorem 3.** *Let Assumptions 1 and 2 be satisfied. Then, Algorithm 1 with the parameters*[2]

$$\eta_t = \frac{R}{\sqrt{2.75 \cdot \sum_{k=1}^{t-1} \|\boldsymbol{g}_k\|_{p^*}^2}} \quad \text{and any} \quad h_t \leq \frac{7R}{200 b_q(d)\sqrt{t}} d^{\frac{1}{2}+\frac{1}{q\wedge 2}-\frac{1}{p}} \ ,$$

*satisfies for any $\boldsymbol{x} \in \Theta$*

$$\mathbf{E}\left[\sum_{t=1}^T \left(f_t(\boldsymbol{x}_t) - f_t(\boldsymbol{x})\right)\right] \leq 110.6 \cdot RL\sqrt{Td^{1+\frac{2}{q\wedge 2}-\frac{2}{p}}} \ .$$

The above result gives, up to an absolute constant, the same convergence rate as that of the non-adaptive Theorem 1. In other words, the price for adaptive algorithm does not depend on the parameters of the problem. Finally, we derive an adaptive algorithm under Assumption 3.

**Theorem 4.** *Let Assumptions 1 and 3 be satisfied. Then, Algorithm 1 with the parameters*

$$\eta_t = \frac{R}{\sqrt{2.75 \cdot \sum_{k=1}^{t-1} \|\boldsymbol{g}_k\|_{p^*}^2}} \quad \text{and any} \quad h_t = \left(6.65\sqrt{6} \cdot \frac{R}{b_q(d)}\right)^{\frac{1}{2}} t^{-\frac{1}{4}} d^{1-\frac{1}{2p}} \ ,$$

*satisfies for any $\boldsymbol{x} \in \Theta$*

$$\mathbf{E}\left[\sum_{t=1}^T \left(f_t(\boldsymbol{x}_t) - f_t(\boldsymbol{x})\right)\right] \leq 110.6 \cdot RL\sqrt{Td^{1+\frac{2}{q\wedge 2}-\frac{2}{p}}}$$

$$+ 5.9 \cdot \sqrt{R} \, (\sigma+L) \, T^{\frac{3}{4}} \cdot \begin{cases} \sqrt{qd^{1+\frac{1}{q}-\frac{1}{p}}} & \text{if } q \in [1, \log(d)) \\ \sqrt{e\log(d)d^{1-\frac{1}{p}}} & \text{if } q \geq \log(d) \end{cases} \ .$$

Note that the bound of Theorem 4 has a less advantageous dependency on $\sigma$ and $L$ compared to Theorem 2, where we had $\sqrt{\sigma L}$ instead of $\sigma + L$. We remark that if $\sigma$ is known but $L$ is unknown, one can recover the $\sqrt{\sigma L}$ dependency by selecting $h_t$ depending on $\sigma$. We do not state this result that can be derived in a similar way and favor here only the fully adaptive version.

## 6    Elements of proofs

In this section, we outline major ingredients for the proofs of Theorems 1 – 4. The full proofs can be found in Appendix C. Here, we only focus on novel elements without reproducing the general scheme of online dual averaging analysis [see e.g., 26, 31]. Namely, we highlight two key facts, which are the smoothing lemma (Lemma 1) and the weighted Poincaré type inequality for the uniform measure on $\partial B_1^d$ (Lemma 3) used to control the variance.

### 6.1    Bias and smoothing lemma

First, as in the prior work that was using smoothing ideas [see e.g., 16, 22, 33], we show that our gradient estimate $\boldsymbol{g}_t$ is an unbiased estimator of a surrogate version of $f_t$ and establish its approximation properties.

**Lemma 1** (Smoothing lemma). *Fix $h > 0$ and $q \in [1, \infty]$. Let $f : \mathbb{R}^d \to \mathbb{R}$ be an L-Lipschitz function w.r.t. the $\ell_q$-norm. Let $\boldsymbol{U}$ be distributed uniformly on $B_1^d$ and $\boldsymbol{\zeta}$ be distributed uniformly on $\partial B_d^1$. Let $f_h(\boldsymbol{x}) \triangleq \mathbf{E}[f(\boldsymbol{x} + h\boldsymbol{U})]$ for $\boldsymbol{x} \in \mathbb{R}^d$. Then $f_h$ is differentiable and*

$$\mathbf{E}\left[\frac{d}{2h}\left(f(\boldsymbol{x} + h\boldsymbol{\zeta}) - f(\boldsymbol{x} - h\boldsymbol{\zeta})\right)\operatorname{sign}(\boldsymbol{\zeta})\right] = \nabla f_h(\boldsymbol{x}) \ .$$

*Furthermore, we have for all $d \geq 3$ and all $\boldsymbol{x} \in \mathbb{R}^d$,*

$$|f_h(\boldsymbol{x}) - f(\boldsymbol{x})| \leq b_q(d)Lh \ . \tag{1}$$

*Finally, if $\Theta \subset \mathbb{R}^d$ is convex, $f$ is convex in $\Theta + hB_1^d$, then $f_h$ is convex in $\Theta$ and $f_h(\boldsymbol{x}) \geq f(\boldsymbol{x})$ for $\boldsymbol{x} \in \Theta$.*

---

[2]We adopt the convention that $\eta_1 = 1$ and $1/0 = 1$ in the definition of $\eta_t$.

*Proof.* There are three claims to prove. For the first one, we notice that $\boldsymbol{\zeta}$ has the same distribution as $-\boldsymbol{\zeta}$, hence,

$$\mathbf{E}\left[\frac{d}{2h}\big(f(\boldsymbol{x}+h\boldsymbol{\zeta})-f(\boldsymbol{x}-h\boldsymbol{\zeta})\big)\operatorname{sign}(\boldsymbol{\zeta})\right]=\mathbf{E}\left[\frac{d}{h}f(\boldsymbol{x}+h\boldsymbol{\zeta})\operatorname{sign}(\boldsymbol{\zeta})\right]\ ,$$

and the first claim follows from Theorem 6 in the Appendix (a version of Stokes', or divergence, theorem) applied to $g(\cdot)=f(\boldsymbol{x}+h\cdot)$ with observation that $\nabla g(\cdot)=h\nabla f(\boldsymbol{x}+h\cdot)$ where $\nabla f$ is the gradient defined almost everywhere and whose existence is ensured by the Rademacher theorem.

We now prove the approximation property (1). Assuming $d\geq 3$ grants that $\log(d)\geq 1$. Since $f$ is $L$-Lipschitz w.r.t. the $\ell_q$-norm we get that, for any $\boldsymbol{x}\in\mathbb{R}^d$,

$$|\mathsf{f}_h(\boldsymbol{x})-f(\boldsymbol{x})|\leq Lh\,\mathbf{E}\|\boldsymbol{U}\|_q\ . \tag{2}$$

If $q\in[1,\log(d))$ then (1) follows from Lemma 2. If $q\geq\log(d)$ then using again Lemma 2 we find

$$\mathbf{E}\|\boldsymbol{U}\|_q\leq\mathbf{E}\|\boldsymbol{U}\|_{\log(d)}\leq\frac{\log(d)d^{\frac{1}{\log(d)}}}{d+1}=\frac{e\log(d)}{d+1}\ ,$$

which together with (2) yields the desired bound.

Finally, if $f$ is convex in $\Theta+hB_1^d$, then for all $\boldsymbol{x},\boldsymbol{x}'\in\Theta$ and $\alpha\in[0,1]$ we have

$$\mathsf{f}_h(\alpha\boldsymbol{x}+(1-\alpha)\boldsymbol{x}')=\mathbf{E}\left[f\big(\alpha(\boldsymbol{x}+h\boldsymbol{U})+(1-\alpha)(\boldsymbol{x}'+h\boldsymbol{U})\big)\right]\leq\alpha\mathsf{f}_h(\boldsymbol{x})+(1-\alpha)\mathsf{f}_h(\boldsymbol{x}')\ .$$

Thus $\mathsf{f}_h$ is indeed convex on $\Theta$. Furthermore, again by convexity of $f$, we deduce that for any $\boldsymbol{x}\in\Theta$

$$\mathsf{f}_h(\boldsymbol{x})=\mathbf{E}[f(\boldsymbol{x}+h\boldsymbol{U})]\geq\mathbf{E}\left[f(\boldsymbol{x})+\langle\boldsymbol{w},h\boldsymbol{U}\rangle\right]=f(\boldsymbol{x})\quad\text{where }\boldsymbol{w}\in\partial f(\boldsymbol{x})\ .\qquad\square$$

The proof of Lemma 1 relies on the control of the $\ell_q$-norm of random vector $\boldsymbol{U}$ established in the next result.

**Lemma 2.** *Let $q\in[1,\infty)$ and let $\boldsymbol{U}$ be distributed uniformly on $B_1^d$. Then $\mathbf{E}\|\boldsymbol{U}\|_q\leq\frac{qd^{\frac{1}{q}}}{d+1}$.*

*Proof.* Let $W_1,\ldots,W_d,W_{d+1}$ be i.i.d. random variables having the Laplace distribution with mean 0 and scale parameter 1. Set $\boldsymbol{W}=(W_1,\ldots,W_d)$. Then, following [6, Theorem 1] we have

$$\boldsymbol{U}\stackrel{d}{=}\frac{\boldsymbol{W}}{\|\boldsymbol{W}\|_1+|W_{d+1}|}\ ,$$

where $\stackrel{d}{=}$ denotes equality in distribution. Furthermore, [30, Lemma 1] states that the random variables

$$\frac{(\boldsymbol{W},|W_{d+1}|)}{\|\boldsymbol{W}\|_1+|W_{d+1}|}\qquad\text{and}\qquad\|\boldsymbol{W}\|_1+|W_{d+1}|\ ,$$

are independent. Hence, for any $q\in[1,\infty)$, it holds that

$$\mathbf{E}\|\boldsymbol{U}\|_q=\frac{\mathbf{E}\|\boldsymbol{W}\|_q}{\mathbf{E}\|(\boldsymbol{W},W_{d+1})\|_1}=\frac{1}{d+1}\mathbf{E}\|\boldsymbol{W}\|_q\stackrel{(a)}{\leq}\frac{1}{d+1}\left(\mathbf{E}\|\boldsymbol{W}\|_q^q\right)^{\frac{1}{q}}=\frac{d^{\frac{1}{q}}\Gamma^{\frac{1}{q}}(q+1)}{d+1}\stackrel{(b)}{\leq}\frac{qd^{\frac{1}{q}}}{d+1}\ ,$$

where $(a)$ follows from Jensen's inequality and $(b)$ uses the fact that $\Gamma^{1/q}(q+1)\leq q$ for $q\geq 1$. $\square$

## 6.2 Variance and weighted Poincaré type inequality

We additionally need to control the squared $\ell_{p^*}$-norm of each gradient estimator $\boldsymbol{g}_t$. This is where we get the main improvement of our procedure compared to previously proposed methods. To derive the result, we first establish the following lemma of independent interest, which allows us to control the variance of Lipschitz functions on $\partial B_1^d$. The proof of this lemma is given in the Appendix.

**Lemma 3.** *Let $d\geq 3$. Assume that $G:\mathbb{R}^d\to\mathbb{R}$ is a continuously differentiable function, and $\boldsymbol{\zeta}$ is distributed uniformly on $\partial B_1^d$. Then*

$$\operatorname{Var}(G(\boldsymbol{\zeta}))\leq\frac{4}{d(d-2)}\mathbf{E}\left[\|\nabla G(\boldsymbol{\zeta})\|_2^2\left(1+\sqrt{d}\|\boldsymbol{\zeta}\|_2\right)^2\right]\ .$$

*Furthermore, if $G:\mathbb{R}^d\to\mathbb{R}$ is an $L$-Lipschitz function w.r.t. the $\ell_2$-norm then*

$$\operatorname{Var}(G(\boldsymbol{\zeta}))\leq\frac{4L^2}{d(d-2)}\left(1+\sqrt{\frac{2d}{d+1}}\right)^2\ .$$

**Remark 1.** *Since $d^2/(d(d-2)) \leq 3$ for all $d \geq 3$, the last inequality of Lemma 3 implies that*

$$\mathrm{Var}(G(\boldsymbol{\zeta})) \leq 12\big(1+\sqrt{2}\big)^2\big(L/d\big)^2\ , \qquad \forall d \geq 3\ . \tag{3}$$

We can now deduce the following bound on the squared $\ell_{p^*}$-norm of $\boldsymbol{g}_t$.

**Lemma 4.** *Let $p \in [1,\infty]$ and $p^* = \frac{p}{p-1}$. Assume that $f_t$ is $L$-Lipschitz w.r.t. the $\ell_q$-norm. Then, for all $d \geq 3$,*

$$\mathbf{E}\|\boldsymbol{g}_t\|_{p^*}^2 \leq 12(1+\sqrt{2})^2 L^2 d^{1+\frac{2}{q\wedge 2}-\frac{2}{p}} + \begin{cases} 0 & \text{under canceling noise Assumption 2,} \\[2mm] \dfrac{d^{4-\frac{2}{p}}\sigma^2}{h^2} & \text{under adversarial noise Assumption 3.} \end{cases}$$

*Proof.* Using the definition of $\boldsymbol{g}_t$ we get

$$\mathbf{E}[\|\boldsymbol{g}_t\|_{p^*}^2 \mid \boldsymbol{x}_t] = \frac{d^2}{4h^2}\mathbf{E}[(f_t(\boldsymbol{x}_t+h\boldsymbol{\zeta}_t)-f_t(\boldsymbol{x}_t-h\boldsymbol{\zeta}_t)+\xi_t'-\xi_t'')^2\|\mathrm{sign}(\boldsymbol{\zeta}_t)\|_{p^*}^2 \mid \boldsymbol{x}_t]$$

$$= \frac{d^{4-\frac{2}{p}}}{4h^2}\mathbf{E}[(f_t(\boldsymbol{x}_t+h\boldsymbol{\zeta}_t)-f_t(\boldsymbol{x}_t-h\boldsymbol{\zeta}_t)+\xi_t'-\xi_t'')^2 \mid \boldsymbol{x}_t]\ .$$

Let $G(\boldsymbol{\zeta}) \triangleq f_t(\boldsymbol{x}_t+h\boldsymbol{\zeta}) - f_t(\boldsymbol{x}_t-h\boldsymbol{\zeta})$. First, observe that $\mathbf{E}[G(\boldsymbol{\zeta}_t) \mid \boldsymbol{x}_t] = 0$ and under both Assumption 2 and Assumption 3(ii) it holds that $\mathbf{E}[G(\boldsymbol{\zeta}_t)(\xi_t'-\xi_t'') \mid \boldsymbol{x}_t] = 0$. Using these remarks and the fact that under adversarial noise Assumption 3, $\mathbf{E}[(\xi_t'-\xi_t'')^2 \mid \boldsymbol{x}_t] \leq 4\sigma^2$, we find:

$$\mathbf{E}[\|\boldsymbol{g}_t\|_{p^*}^2 \mid \boldsymbol{x}_t] \leq \frac{d^{4-\frac{2}{p}}}{4h^2}\left(\mathrm{Var}(G(\boldsymbol{\zeta}_t) \mid \boldsymbol{x}_t) + \begin{cases} 0 & \text{under cancelling noise Assumption 2} \\ 4\sigma^2 & \text{under adversarial noise Assumption 3} \end{cases}\right).$$

Furthermore, since $f_t$ is $L$-Lipschitz, w.r.t. the $\ell_q$-norm, the map $\boldsymbol{\zeta} \mapsto G(\boldsymbol{\zeta})$ is $\big(2Lhd^{\frac{1}{q\wedge 2}-\frac{1}{2}}\big)$-Lipschitz w.r.t. the $\ell_2$-norm. Applying (3) to bound $\mathrm{Var}(G(\boldsymbol{\zeta}_t) \mid \boldsymbol{x}_t)$, yields the desired result. $\square$

Note that under adversarial noise Assumption 3, the bound on squared $\ell_{p^*}$-norm of $\boldsymbol{g}_t$ gets an additional term $d^{4-\frac{2}{p}}\sigma^2 h^{-2}$. In contrast to the case of canceling noise Assumption 2, this does not allow us to take $h$ arbitrary small hence inducing the bias-variance trade-off.

## 7 Numerical illustration

In this section, we provide a numerical comparison of our algorithm with the method based on $\ell_2$-randomization from Shamir [33] (see Appendix D for the definition). We consider the no noise model and $f_t = f, \forall t$, with the function $f : \mathbb{R}^d \to \mathbb{R}$ defined as

$$f(\boldsymbol{x}) = \|\boldsymbol{x}-\boldsymbol{c}\|_2 + \|\boldsymbol{x}-0.1\cdot\boldsymbol{c}\|_1\ ,$$

where $\boldsymbol{c} = (c_1,\ldots,c_d)^\top \in \mathbb{R}^d$ such that $c_j = \exp(j)/\sum_{i=1}^d \exp(i)$ for $j = 1,\ldots,d$. We choose

$$\Theta = \big\{\boldsymbol{x} \in \mathbb{R}^d\ :\ \|\boldsymbol{x}\|_1 = 1,\ \boldsymbol{x} \geq 0\big\} \quad \text{and} \quad V(\boldsymbol{x}) = \sum_{j=1}^d x_j \log(x_j)\ .$$

As stated in Example 2, $V$ is 1-strongly convex on $\Theta$ w.r.t. the $\ell_1$-norm and $R \leq \sqrt{\log(d)}$. Moreover, $f$ is a Lipschitz function w.r.t. the $\ell_1$-norm. We deploy the adaptive parameterization that appears in Theorem 3. In Figure 1 we present the optimization error of the algorithms, which is defined as

$$f\left(\frac{1}{t}\sum_{i=1}^t \boldsymbol{x}_i\right) - \min_{\boldsymbol{x}\in\Theta} f(\boldsymbol{x})\ .$$

The results are reported over 30 trials. We plot all the 30 runs alongside the average performance. One can observe that the $\ell_1$-randomization method behaves significantly better than the $\ell_2$-randomization algorithm. The theoretical bound for our method in this setup has a $\sqrt{\log d}$ gain in the rate.

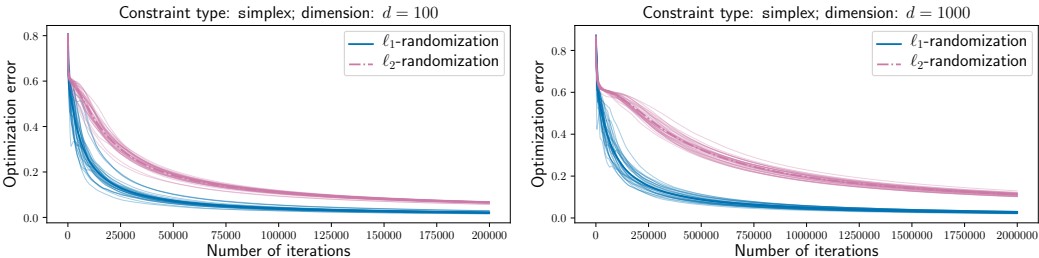

Figure 1: Opt. error vs. number of iterations for $\ell_2$-randomization (as in [33]) and our method.

## 8  Discussion and comparison to prior work

We introduced and analyzed a novel estimator for the gradient based on randomization over the $\ell_1$-sphere. We established guarantees for the online dual averaging algorithm with the gradient replaced by the proposed estimator. We provided an anytime and completely data-driven algorithm, which is adaptive to all parameters of the problem. Our analysis is based on deriving a weighted Poincaré type inequality for the uniform measure on the $\ell_1$-sphere that may be of independent interest. Under the *canceling noise assumption* and $q \in \{1, 2\}$, our setting is analogous to [14, 33]. For the case $q = p = 2$ and canceling noise, we show that the performance of our method is the same as in [33, Corollary 2] up to absolute constants that were not made explicit in [33]. For the case of $q = p = 1$ and canceling noise, we improved the bound [33, Corollary 3] by a $\sqrt{\log(d)}$ factor. For the case $q = 2$, $p \geq 1$, comparing with the lower bound in [14, Proposition 1], shows that the result of Theorem 1 is minimax optimal. For the case $q = p = 1$, [14, Proposition 2] shows that our result in Theorem 1 is optimal up to a $\log(d)$ factor.

Under the *adversarial noise assumption*, Theorem 2 provides the rate $O(T^{3/4})$, that is, we get an additional $T^{1/4}$ factor compared to the canceling noise case. It remains unclear whether it is optimal under adversarial noise – this question deserves further investigation. Note that, under sub-Gaussian i.i.d. noise assumption and $q = p = 2$, one can achieve the rate $\tilde{O}(d^a \sqrt{T})$ with a relatively big $a > 0$ [2, 8, 13, 21]. In particular, with an ellipsoid type method [21] obtains the rate $\mathcal{O}(d^{4.5} \sqrt{T} \log(T)^2)$ for the cumulative regret.

Finally, let us discuss the compactness of $\Theta$. It is straightforward to extend the results of Theorems 1, 2 to any closed convex $\Theta$ considering the regret against a fixed action $x \in \Theta$. Indeed, using [26, Corollary 7.9], one only needs to replace $R$ appearing in both Theorems 1, 2 by an upper bound on $\sqrt{V(x) - \inf_{x' \in \Theta} V(x')}$. The adaptive case is more complicated. One way to tackle this case is to use [27, Theorem 1] requiring a control of $\mathbf{E} \max_{t=1,\dots,T} \|g_t\|_{p^*}$. This term can be controlled under the canceling noise Assumption 2 using the Lipschitzness of $f_t$'s, so that Theorem 3 extends to unbounded $\Theta$. However, without the canceling noise assumption, following the approach outlined above, one needs to control $\mathbf{E} \max_{t=1,\dots,T} \frac{|\xi'_t - \xi''_t|}{h_t}$. The adversarial noise Assumption 3 is not sufficient to reasonably control this term, so that extending Theorem 4 to unbounded $\Theta$ is not possible without further assumptions.

## Acknowledgments and disclosure of funding

The work of E. Chzhen was supported by ANR PIA funding: ANR-20-IDEES-0002. The work of M. Pontil is partially supported by European Union's Horizon 2020 research and innovation programme under ELISE grant agreement No 951847. The research of A. Akhavan and A. B. Tsybakov is supported by a grant of the French National Research Agency (ANR), "Investissements d'Avenir" (LabEx Ecodec/ANR-11-LABX-0047).

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
