# Supplementary Material

This supplementary material contains the proofs and results omitted from the main body. In Appendix A we recall the appropriate version of the Stokes' theorem and discuss its applicability for Lipschitz functions on $B_1^d$. In Appendix B we provide the proof of Lemma 3. Finally, in Appendix C we provide the proofs of Theorems 1, 2, 3, 4.

**Additional notation** For two functions $g, \eta : \mathbb{R}^d \to \mathbb{R}$, we denote by $\eta \star g$ their convolution defined point-wise for $\boldsymbol{x} \in \mathbb{R}^d$ as

$$\big(\eta \star g\big)(\boldsymbol{x}) = \int_{\mathbb{R}^d} \eta(\boldsymbol{x} - \boldsymbol{x}')g(\boldsymbol{x}') \, \mathrm{d}\boldsymbol{x}' \ .$$

The standard mollifier $\eta_\epsilon : \mathbb{R}^d \to \mathbb{R}$ is defined as $\eta_\epsilon(\boldsymbol{x}) = \epsilon^{-d}\eta_1(\boldsymbol{x}/\epsilon)$ for $\epsilon > 0$ and $\boldsymbol{x} \in \mathbb{R}$, where $\eta_1 : \mathbb{R}^d \to \mathbb{R}$ is defined as

$$\eta_1(\boldsymbol{x}) = \begin{cases} C \exp\left(\frac{1}{\|\boldsymbol{x}\|_2^2 - 1}\right) & \text{if } \|\boldsymbol{x}\|_2 \leq 1 \\ 0 & \text{otherwise} \end{cases} \ ,$$

with $C$ chosen so that $\int_{\mathbb{R}^d} \eta_1(\boldsymbol{x}) \, \mathrm{d}\boldsymbol{x} = 1$.

## A    Integration by parts

We first recall the following result that can be found in [34, Section 13.3.5, Exercise 14a].

**Theorem 5** (Integration by parts in a multiple integral). *Let $D$ be an open connected subset of $\mathbb{R}^d$ with a piecewise smooth boundary $\partial D$ oriented by the outward unit normal $\boldsymbol{n} = (n_1, \ldots, n_d)^\top$. Let $g$ be a continuously differentiable function in $D \cup \partial D$. Then*

$$\int_D \nabla g(\boldsymbol{u}) \, \mathrm{d}\boldsymbol{u} = \int_{\partial D} g(\boldsymbol{\zeta})\boldsymbol{n}(\boldsymbol{\zeta}) \, \mathrm{d}S(\boldsymbol{\zeta}) \ .$$

**Remark 2.** *We refer to [34, Section 12.3.2, Definitions 4 and 5] for the definition of piecewise smooth surfaces and their orientations respectively.*

The idea of using the instance of Theorem 5 (also called Stokes' theorem) with $D = B_2^d$ to obtain $\ell_2$-randomized estimators of the gradient belongs to Nemirovsky and Yudin [22]. It was further used in several papers [5, 16, 31, 33] to mention just a few. Those papers were referring to [22] but [22] did not provide an exact statement of the result (nor a reference) and only tossed the idea in a discussion. However, the classical analysis formulation as presented in Theorem 5 does not apply to Lipschitz continuous functions that were considered in [5, 16, 31, 33]. We are not aware of whether its extension to Lipschitz continuous functions, though rather standard, is proved in the literature.

In this paper, we apply Theorem 5 with the $\ell_1$-ball $D = B_1^d$. Our aim in this section is to provide a variant of Theorem 5 applicable to a Lipschitz continuous function $g : \mathbb{R}^d \to \mathbb{R}$, which is not necessarily continuously differentiable on $D \cup \partial D = B_1^d \cup \partial B_1^d$. To this end, we will go through the argument of approximating $g$ by $C^\infty(\Omega)$ functions, where $\Omega \subset \mathbb{R}^d$ is an open bounded connected subset of $\mathbb{R}^d$ such that $D \cup \partial D \subset \Omega$. Let $g_n = \eta_{1/n} \star g$, where $\eta_{1/n}$ is the standard mollifier. Let $g : \mathbb{R}^d \to \mathbb{R}$ be a function satisfying the Lipschitz condition w.r.t. the $\ell_1$-norm: $|g(\boldsymbol{u}) - g(\boldsymbol{u}')| \leq L\|\boldsymbol{u} - \boldsymbol{u}'\|_1$. Since $g$ is continuous in $\Omega$ and, by construction $D \cup \partial D \subset \Omega$, then using basic properties of mollification [see e.g., 15, Theorem 4.1 (ii)] we have

$$g_n \longrightarrow g$$

uniformly on $D \cup \partial D$ (in particular, uniformly on $\partial D$). Furthermore, let $\nabla g$ be the gradient of $g$, which by Rademacher theorem [see e.g., 15, Theorem 3.2] is well defined almost everywhere w.r.t. the Lebesgue measure and

$$\|\nabla g(\boldsymbol{u})\|_\infty \leq L \qquad \text{a.e.}$$

It follows that $\frac{\partial g}{\partial u_j}$ is absolutely integrable on $\Omega$ for any $j \in [d]$. Furthermore, since

$$\frac{\partial g_n}{\partial u_j} = \eta_{1/n} \star \left( \frac{\partial g}{\partial u_j} \right) \quad,$$

we can apply [15, Theorem 4.1 (iii)] that yields

$$\int_D \|\nabla g_n(\boldsymbol{u}) - \nabla g(\boldsymbol{u})\|_2 \, \mathrm{d}\boldsymbol{u} \longrightarrow 0 \quad.$$

Combining the above remarks we obtain that the result of Theorem 5 is valid for functions $g$ that are Lipschitz continuous w.r.t. the $\ell_1$-norm. Thus, it is also valid when the Lipschitz condition is imposed w.r.t. any $\ell_q$-norm with $q \in [1, \infty]$. Specifying this conclusion for the particular case $D = B_1^d$, we obtain the following theorem.

**Theorem 6.** *Let the function $g : \mathbb{R}^d \to \mathbb{R}$ be Lipschitz continuous w.r.t. the $\ell_q$-norm with $q \in [1, \infty]$. Then*

$$\int_{B_1^d} \nabla g(\boldsymbol{u}) \, \mathrm{d}\boldsymbol{u} = \frac{1}{\sqrt{d}} \int_{\partial B_1^d} g(\boldsymbol{\zeta}) \operatorname{sign}(\boldsymbol{\zeta}) \, \mathrm{d}S(\boldsymbol{\zeta}) \quad,$$

*where $\nabla g(\cdot)$ is defined up to a set of zero Lebesgue measure by the Rademacher theorem.*

# B   Proof of Lemma 3

To prove Lemma 3, we first recall the weighted Poincaré inequality for the univariate exponential measure (mean 0 and scale parameter 1 Laplace distribution).

**Lemma 5** (Lemma 2.1 in [9]). *Let $W$ be mean 0 and scale parameter 1 Laplace random variable. Let $g : \mathbb{R} \to \mathbb{R}$ be continuous almost everywhere differentiable function such that*

$$\mathbf{E}[|g(W)|] < \infty \quad and \quad \mathbf{E}[|g'(W)|] < \infty \quad and \quad \lim_{|w| \to \infty} g(w) \exp(-|w|) = 0 \quad,$$

*then,*

$$\mathbf{E}[(g(W) - \mathbf{E}[g(W)])^2] \leq 4\mathbf{E}[(g'(W))^2].$$

We are now in a position to prove Lemma 3. The proof is inspired by [7, Lemma 2].

*Proof of Lemma 3.* Throughout the proof, we assume without loss of generality that $\mathbf{E}[G(\boldsymbol{\zeta})] = 0$. Indeed, if it is not the case, we use the result for the centered function $\tilde{G}(\boldsymbol{\zeta}) = G(\boldsymbol{\zeta}) - \mathbf{E}[G(\boldsymbol{\zeta})]$, which has the same gradient.

First, consider the case of continuously differentiable $G$. Let $\boldsymbol{W} = (W_1, \ldots, W_d)$ be a vector of i.i.d. mean 0 and scale parameter 1 Laplace random variables and define $\boldsymbol{T}(\boldsymbol{w}) = \boldsymbol{w}/\|\boldsymbol{w}\|_1$. Introduce the notation

$$F(\boldsymbol{w}) \triangleq \|\boldsymbol{w}\|_1^{1/2} G(\boldsymbol{T}(\boldsymbol{w})) \quad.$$

Lemma 1 in [30] asserts that, for $\boldsymbol{\zeta}$ uniformly distributed on $\partial B_1^d$,

$$\boldsymbol{T}(\boldsymbol{W}) \overset{d}{=} \boldsymbol{\zeta} \quad and \quad \boldsymbol{T}(\boldsymbol{W}) \text{ is independent of } \|\boldsymbol{W}\|_1 \quad. \tag{4}$$

In particular,

$$\operatorname{Var}(F(\boldsymbol{W})) = d \operatorname{Var}(G(\boldsymbol{\zeta})) \quad.$$

Using the Efron-Stein inequality [see e.g., 11, Theorem 3.1] we obtain

$$\operatorname{Var}(F(\boldsymbol{W})) \leq \sum_{i=1}^d \mathbf{E}\left[ \operatorname{Var}_i(F) \right] \quad,$$

where

$$\operatorname{Var}_i(F) = \mathbf{E}\left[ \left( F(\boldsymbol{W}) - \mathbf{E}[F(\boldsymbol{W}) \mid \boldsymbol{W}^{-i}] \right)^2 \mid \boldsymbol{W}^{-i} \right]$$

with $\boldsymbol{W}^{-i} \triangleq (W_1, \ldots, W_{i-1}, W_{i+1}, \ldots, W_d)$. Note that on the event $\{\boldsymbol{W}^{-i} \neq \boldsymbol{0}\}$ (whose complement has zero measure), the function

$$w \mapsto F(W_1, \ldots, W_{i-1}, w, W_{i+1}, \ldots, W_d) \ ,$$

satisfies the assumptions of Lemma 5. Thus,

$$d \operatorname{Var}(G(\boldsymbol{\zeta})) = \operatorname{Var}(F(\boldsymbol{W})) \leq 4 \sum_{j=1}^{d} \mathbf{E}\left[\left(\frac{\partial F}{\partial w_j}(\boldsymbol{W})\right)^2\right] = 4\mathbf{E}\|\nabla F(\boldsymbol{W})\|_2^2 \ . \tag{5}$$

In order to compute $\nabla F(\boldsymbol{W})$, we observe that for every $i \neq j \in [d]$ we have for all $\boldsymbol{w} \neq \boldsymbol{0}$ such that $w_i, w_j \neq 0$

$$\frac{\partial T_i}{\partial w_j}(\boldsymbol{w}) = -\frac{w_i \operatorname{sign}(w_j)}{\|\boldsymbol{w}\|_1^2} \quad \text{and} \quad \frac{\partial T_i}{\partial w_i}(\boldsymbol{w}) = \frac{1}{\|\boldsymbol{w}\|_1} - \frac{w_i \operatorname{sign}(w_i)}{\|\boldsymbol{w}\|_1^2} \ .$$

Thus, the Jacobi matrix of $\boldsymbol{T}(\boldsymbol{w})$ has the form

$$\mathbf{J}_{\boldsymbol{T}}(\boldsymbol{w}) = \frac{\mathbf{I}}{\|\boldsymbol{w}\|_1} - \frac{\boldsymbol{w}(\operatorname{sign}(\boldsymbol{w}))^\top}{\|\boldsymbol{w}\|_1^2} = \frac{1}{\|\boldsymbol{w}\|_1}\left(\mathbf{I} - \boldsymbol{T}(\boldsymbol{w})\big(\operatorname{sign}(\boldsymbol{w})\big)^\top\right) \ .$$

It follows that almost surely

$$\nabla F(\boldsymbol{W}) = \frac{1}{2\|\boldsymbol{W}\|_1^{1/2}}G(\boldsymbol{T}(\boldsymbol{W}))\operatorname{sign}(\boldsymbol{W}) + \frac{1}{\|\boldsymbol{W}\|_1^{1/2}}\left(\mathbf{I} - \boldsymbol{T}(\boldsymbol{W})\big(\operatorname{sign}(\boldsymbol{W})\big)^\top\right)\nabla G(\boldsymbol{T}(\boldsymbol{W})) \ .$$

Observe that since $\langle \operatorname{sign}(\boldsymbol{W}), \boldsymbol{T}(\boldsymbol{W})\rangle = 1$ almost surely, we have

$$\big(\operatorname{sign}(\boldsymbol{W})\big)^\top\left(\mathbf{I} - \boldsymbol{T}(\boldsymbol{W})\big(\operatorname{sign}(\boldsymbol{W})\big)^\top\right)\nabla G(\boldsymbol{T}(\boldsymbol{W})) = 0 \quad \text{almost surely} \ .$$

The above two equations imply that almost surely

$$4\|\nabla F(\boldsymbol{W})\|_2^2 = \frac{d}{\|\boldsymbol{W}\|_1}G^2(\boldsymbol{T}(\boldsymbol{W})) + \frac{4}{\|\boldsymbol{W}\|_1}\left\|\left(\mathbf{I} - \boldsymbol{T}(\boldsymbol{W})\big(\operatorname{sign}(\boldsymbol{W})\big)^\top\right)\nabla G(\boldsymbol{T}(\boldsymbol{W}))\right\|_2^2$$

$$\leq \frac{d}{\|\boldsymbol{W}\|_1}G^2(\boldsymbol{T}(\boldsymbol{W})) + \frac{4}{\|\boldsymbol{W}\|_1}\|\nabla G(\boldsymbol{T}(\boldsymbol{W}))\|_2^2\left(1 + \sqrt{d}\|\boldsymbol{T}(\boldsymbol{W})\|_2\right)^2 \ ,$$

where we used the fact that the operator norm of $\mathbf{I} - \boldsymbol{a}\boldsymbol{b}^\top$ is not greater than $1 + \|\boldsymbol{a}\|_2\|\boldsymbol{b}\|_2$. Combining the above bound with (5), and using the facts that $\mathbf{E}[\|\boldsymbol{W}\|_1^{-1}] = \frac{1}{d-1}$, $\mathbf{E}[G(\boldsymbol{T}(\boldsymbol{W}))] = \mathbf{E}[G(\boldsymbol{\zeta})] = 0$ and the independence of $\|\boldsymbol{W}\|_1$ and $\boldsymbol{T}(\boldsymbol{W})$ (cf. (4)) yields

$$d\left(1 - \frac{1}{d-1}\right)\operatorname{Var}(G(\boldsymbol{\zeta})) \leq \frac{4}{d-1}\mathbf{E}\left[\|\nabla G(\boldsymbol{T}(\boldsymbol{W}))\|_2^2(1 + \sqrt{d}\|\boldsymbol{T}(\boldsymbol{W})\|_2)^2\right] \ .$$

Rearranging, we deduce the first claim of the lemma since $\boldsymbol{T}(\boldsymbol{W}) \stackrel{d}{=} \boldsymbol{\zeta}$.

To prove the second statement of the lemma regarding Lipschitz functions, it is sufficient to apply the first one to $G_n$—the sequence of smoothed versions of $G$ such that $G_n \in C^\infty(\mathbb{R})$ and

$$G_n \longrightarrow G \ ,$$

uniformly on every compact subset, and $\sup_{n\geq 1}\|\nabla G_n(\boldsymbol{x})\|_2 \leq L$ for almost all $\boldsymbol{x} \in \mathbb{R}^d$. A sequence $G_n$ satisfying these properties can be constructed by standard mollification due to the fact that $G$ is Lipschitz continuous [see e.g., 15, Theorem 4.2]. Finally, to obtain the value $\mathbf{E}\|\boldsymbol{T}(\boldsymbol{W})\|_2^2 = \mathbf{E}\|\boldsymbol{\zeta}\|_2^2$ we use Lemma 6 below. $\qquad\square$

**Lemma 6.** *Let $\boldsymbol{\zeta}$ be distributed uniformly on $\partial B_1^d$. Then, $\mathbf{E}\|\boldsymbol{\zeta}\|_2^2 = \frac{2}{d+1}$.*

*Proof.* We use the same tools as in the proof of Lemma 2. Let $\boldsymbol{W} = (W_1, \ldots, W_d)$ be a vector of i.i.d. random variables following the Laplace distribution with mean 0 and scale parameter 1. By (4) we have that $\boldsymbol{\zeta} \stackrel{d}{=} \frac{\boldsymbol{W}}{\|\boldsymbol{W}\|_1}$ and $\boldsymbol{\zeta}$ is independent of $\|\boldsymbol{W}\|_1$. Therefore,

$$\mathbf{E}\|\boldsymbol{\zeta}\|_2^2 = \frac{\mathbf{E}\|\boldsymbol{W}\|_2^2}{\mathbf{E}\|\boldsymbol{W}\|_1^2} \ . \tag{6}$$

Here,

$$\mathbf{E}\|\boldsymbol{W}\|_2^2 = \sum_{j=1}^d \mathbf{E}[W_j^2] = d\mathbf{E}[W_1^2] = 2d \ . \tag{7}$$

Furthermore, $\|\boldsymbol{W}\|_1$ follows the Erlang distribution with parameters $(d, 1)$, which implies

$$\mathbf{E}\|\boldsymbol{W}\|_1^2 = \frac{1}{\Gamma(d)} \int_0^\infty x^{d+1} \exp(-x)\, \mathrm{d}x = \frac{\Gamma(d+2)}{\Gamma(d)} \ . \tag{8}$$

The lemma follows by combining (6) – (8). $\qquad\square$

## C Upper bounds

The proofs of Theorems 1, 2, 3, 4 resemble each other. They only differ in the ways of handling the variance terms depending on $\|\boldsymbol{g}_t\|_{p^*}^2$ and in the choice of parameters. For this reason, we suggest the interested reader to follow the proofs in a linear manner starting from the next paragraph.

**Common part of the proofs of Theorems 1, 2.** We start with the part of the proofs that is common for Theorems 1, 2. Fix some $\boldsymbol{x} \in \Theta$. Due to Assumption 1, we can use Lemma 1, which implies

$$\mathbf{E}\left[\sum_{t=1}^T \langle \mathbf{E}\left[\boldsymbol{g}_t \mid \boldsymbol{x}_t\right], \boldsymbol{x}_t - \boldsymbol{x}\rangle\right] = \mathbf{E}\left[\sum_{t=1}^T \langle \nabla\mathsf{f}_{t,h}(\boldsymbol{x}_t), \boldsymbol{x}_t - \boldsymbol{x}\rangle\right] \geq \mathbf{E}\left[\sum_{t=1}^T \left(\mathsf{f}_{t,h}(\boldsymbol{x}_t) - \mathsf{f}_{t,h}(\boldsymbol{x})\right)\right],$$

where $\mathsf{f}_{t,h}(\boldsymbol{x}) = \mathbf{E}[f_t(\boldsymbol{x} + h\boldsymbol{U})]$ with $\boldsymbol{U}$ uniformly distributed on $B_1^d$. Furthermore, by the approximation property derived in Lemma 1 and the standard bound on the cumulative regret of dual averaging algorithm [see e.g., 26, Corollary 7.9.] we deduce that

$$\mathbf{E}\left[\sum_{t=1}^T \left(f_t(\boldsymbol{x}_t) - f_t(\boldsymbol{x})\right)\right] \leq \mathbf{E}\left[\sum_{t=1}^T \langle \mathbf{E}\left[\boldsymbol{g}_t | \boldsymbol{x}_t\right], \boldsymbol{x}_t - \boldsymbol{x}\rangle\right] + L\mathsf{b}_q(d)\sum_{t=1}^T h_t \tag{9}$$
$$\leq \frac{R^2}{\eta} + \frac{\eta}{2}\sum_{t=1}^T \mathbf{E}\|\boldsymbol{g}_t\|_{p^*}^2 + L\mathsf{b}_q(d)\sum_{t=1}^T h_t \ ,$$

where in the last inequality we used the identity $\eta_1 = \ldots = \eta_T = \eta$. The results of Theorems 1, 2 follow from the bound (9) as detailed below.

*Proof of Theorem 1.* Here $h_1 = \ldots = h_T = h$, and we work under Assumption 2. In this case, bounding $\mathbf{E}\|\boldsymbol{g}_t\|_{p^*}$ in (9) via Lemma 4 yields

$$\mathbf{E}\left[\sum_{t=1}^T \left(f_t(\boldsymbol{x}_t) - f_t(\boldsymbol{x})\right)\right] \leq \frac{R^2}{\eta} + 6(1+\sqrt{2})^2 L^2 \cdot \eta T d^{1 + \frac{2}{q \wedge 2} - \frac{2}{p}} + LhT\mathsf{b}_q(d) \ .$$

Minimizing the the right hand side of the above inequality over $\eta > 0$ and substituting $\eta = \frac{R}{L(\sqrt{6}+\sqrt{12})}\sqrt{\frac{d^{-1 - \frac{2}{q \wedge 2} + \frac{2}{p}}}{T}}$ we deduce that

$$\mathbf{E}\left[\sum_{t=1}^T \left(f_t(\boldsymbol{x}_t) - f_t(\boldsymbol{x})\right)\right] \leq 2\left(\sqrt{6} + \sqrt{12}\right) RL d^{\frac{1}{2} + \frac{1}{q \wedge 2} - \frac{1}{p}}\sqrt{T} + LhT\mathsf{b}_q(d) \ .$$

Taking $h \leq \frac{7R}{100\mathsf{b}_q(d)\sqrt{T}}d^{\frac{1}{2} + \frac{1}{q \wedge 2} - \frac{1}{p}}$ makes negligible the second summand in the above bound. This concludes the proof. $\qquad\square$

*Proof of Theorem 2.* Here again $h_1 = \ldots = h_T = h$, but we work under Assumption 3. Then, bounding $\mathbf{E}\|\boldsymbol{g}_t\|_{p^*}$ in (9) via Lemma 4 yields

$$\mathbf{E}\left[\sum_{t=1}^T \left(f_t(\boldsymbol{x}_t) - f_t(\boldsymbol{x})\right)\right] \leq \frac{R^2}{\eta} + \eta T\left(\frac{d^{4 - \frac{2}{p}}\sigma^2}{h^2} + 6\left(1+\sqrt{2}\right)^2 L^2 d^{1 + \frac{2}{q \wedge 2} - \frac{2}{p}}\right) + LhT\mathsf{b}_q(d) \ .$$

Minimizing the right hand side of the above inequality over $\eta > 0$ and substituting the optimal value

$$\eta = \frac{R}{\sqrt{T}} \left( \frac{d^{4-\frac{2}{p}}\sigma^2}{2h^2} + 6\left(1+\sqrt{2}\right)^2 L^2 d^{1+\frac{2}{q\wedge 2}-\frac{2}{p}} \right)^{-\frac{1}{2}} ,$$

results in the following upper bound on the regret

$$\mathbf{E}\left[\sum_{t=1}^{T}\left(f_t(\boldsymbol{x}_t)-f_t(\boldsymbol{x})\right)\right] \leq 2R\sqrt{T}\left( \frac{d^{4-\frac{2}{p}}\sigma^2}{2h^2} + 6\left(1+\sqrt{2}\right)^2 L^2 d^{1+\frac{2}{q\wedge 2}-\frac{2}{p}} \right)^{\frac{1}{2}} + LhT\mathrm{b}_q(d)$$

$$\leq 2\left(\sqrt{6}+\sqrt{12}\right)RL\sqrt{Td^{1+\frac{2}{q\wedge 2}-\frac{2}{p}}} + \sqrt{2}R\sqrt{T}\frac{d^{2-\frac{1}{p}}\sigma}{h} + LhT\mathrm{b}_q(d) ,$$

where for the last inequality we used the fact that $\sqrt{a+b} \leq \sqrt{a} + \sqrt{b}$ for $a, b \geq 0$. Minimizing over $h > 0$ the last expression and substituting the optimal value $h = \left(\frac{\sqrt{2}R\sigma}{L\mathrm{b}_q(d)}\right)^{\frac{1}{2}} T^{-\frac{1}{4}} d^{1-\frac{1}{2p}}$ we get

$$\mathbf{E}\left[\sum_{t=1}^{T}\left(f_t(\boldsymbol{x}_t)-f_t(\boldsymbol{x})\right)\right] \leq 11.9RL\sqrt{Td^{1+\frac{2}{q\wedge 2}-\frac{2}{p}}} + 2.4\sqrt{RL\sigma}T^{\frac{3}{4}}\sqrt{\mathrm{b}_q(d)}d^{\frac{1}{2}-\frac{1}{2p}}. \quad \square$$

**Common part of the proofs of Theorems 3, 4.** Here, we state the common parts of the proofs for Theorems 3, 4. Similar to the first inequality in (9), we have

$$\mathbf{E}\left[\sum_{t=1}^{T}\left(f_t(\boldsymbol{x}_t)-f_t(\boldsymbol{x})\right)\right] \leq \mathbf{E}\left[\sum_{t=1}^{T}\langle\boldsymbol{g}_t, \boldsymbol{x}_t-\boldsymbol{x}\rangle\right] + L\mathrm{b}_q(d)\sum_{t=1}^{T}h_t .$$

Note that without loss of generality, we can assume that $\sum_{k=1}^{t}\|\boldsymbol{g}_k\|_{p^*}^2 \neq 0$, for all $t \geq 1$. This is a consequence of the fact that if $\sum_{k=1}^{t}\|\boldsymbol{g}_k\|_{p^*}^2 = 0$, then the first term on the r.h.s. of the above inequality will be zero up to round $t$. Thus, we can erase these iterates from the cumulative regret, only paying the bias term for those rounds. In what follows we essentially use [27, Corollary 1], which we re-derive for the sake of clarity. Assume that $\eta_t = \frac{\lambda}{\sqrt{\sum_{k=1}^{t-1}\|\boldsymbol{g}_k\|_{p^*}^2}}$ for $t \in \{2, \ldots, T\}$ and $\lambda > 0$. Then, applying [27, Theorem 1] we deduce that

$$\mathbf{E}\left[\sum_{t=1}^{T}\left(f_t(\boldsymbol{x}_t)-f_t(\boldsymbol{x})\right)\right] \leq \left(\frac{R^2}{\lambda} + 2.75 \cdot \lambda\right)\mathbf{E}\left[\sqrt{\sum_{t=1}^{T}\|\boldsymbol{g}_t\|_{p^*}^2}\right]$$

$$+ 3.5D \cdot \mathbf{E}[\max_{t\in[T]}\|\boldsymbol{g}_t\|_{p^*}] + L\mathrm{b}_q(d)\sum_{t=1}^{T}h_t ,$$

where we introduced $D = \sup_{\boldsymbol{u},\boldsymbol{w}\in\Theta}\|\boldsymbol{u}-\boldsymbol{w}\|_p$. By [27, Proposition 1], we have $D \leq \sqrt{8}R$. Moreover, by Jensen's inequality, using the rough bound $\mathbf{E}[\max_{t\in[T]}\|\boldsymbol{g}_t\|_{p^*}] \leq \sqrt{\sum_{t=1}^{T}\mathbf{E}\left[\|\boldsymbol{g}_t\|_{p^*}^2\right]}$, and substituting $\lambda = \frac{R}{\sqrt{2.75}}$, we deduce that

$$\mathbf{E}\left[\sum_{t=1}^{T}\left(f_t(\boldsymbol{x}_t)-f_t(\boldsymbol{x})\right)\right] \leq \left(2\sqrt{2.75}+3.5\sqrt{8}\right)R\sqrt{\sum_{t=1}^{T}\mathbf{E}\left[\|\boldsymbol{g}_t\|_{p^*}^2\right]} + L\mathrm{b}_q(d)\sum_{t=1}^{T}h_t . \quad (10)$$

Proofs of Theorems 3, 4 provided below follow from the above inequality by properly selecting $h_t > 0$.

*Proof of Theorem 3.* The bound of Lemma 4 under Assumption 2 applied to (10) yields

$$\mathbf{E}\left[\sum_{t=1}^{T}\left(f_t(\boldsymbol{x}_t)-f_t(\boldsymbol{x})\right)\right] \leq 2\left(2\sqrt{2.75}+3.5\sqrt{8}\right)\left(\sqrt{3}+\sqrt{6}\right)RL\sqrt{Td^{1+\frac{2}{q\wedge 2}-\frac{2}{p}}} + L\mathrm{b}_q(d)\sum_{t=1}^{T}h_t$$

$$\leq 110.53 \cdot RL\sqrt{Td^{1+\frac{2}{q\wedge 2}-\frac{2}{p}}} + L\mathrm{b}_q(d)\sum_{t=1}^{T}h_t .$$

Taking $h_t \leq \frac{7R}{200\mathrm{b}_q(d)\sqrt{t}} d^{\frac{1}{2}+\frac{1}{q\wedge 2}-\frac{1}{p}}$ makes negligible the last summand in the above bound. This concludes the proof. $\qquad\square$

*Proof of Theorem 4.* Using (10), the bound of Lemma 4 under Assumption 3 and the fact that $\sqrt{a+b} \leq \sqrt{a} + \sqrt{b}$ for $a, b \geq 0$, we deduce that

$$
\mathbf{E}\left[\sum_{t=1}^{T}\left(f_t(\boldsymbol{x}_t) - f_t(\boldsymbol{x})\right)\right] \leq \left(2\sqrt{2.75} + 3.5\sqrt{8}\right) R \left(\sum_{t=1}^{T} \frac{d^{4-\frac{2}{p}}\sigma^2}{h_t^2} + 12(1+\sqrt{2})^2 L^2 T \cdot d^{1+\frac{2}{q\wedge 2}-\frac{2}{p}}\right)^{\frac{1}{2}}
$$
$$
+ L\mathrm{b}_q(d)\sum_{t=1}^{T} h_t
$$
$$
\leq 110.6 \cdot RL\sqrt{Td^{1+\frac{2}{q\wedge 2}-\frac{2}{p}}} + 13.3R \cdot d^{2-\frac{1}{p}}\sigma \left(\sum_{t=1}^{T} \frac{1}{h_t^2}\right)^{\frac{1}{2}}
$$
$$
+ L\mathrm{b}_q(d)\sum_{t=1}^{T} h_t \ .
$$

Since $h_t = \left(6.65\sqrt{6} \cdot \frac{R}{\mathrm{b}_q(d)}\right)^{\frac{1}{2}} t^{-\frac{1}{4}} d^{1-\frac{1}{2p}}$ and $\sum_{t=1}^{T} t^{\frac{1}{2}} \leq \frac{2}{3} T^{\frac{3}{2}}$ and $\sum_{t=1}^{T} t^{-\frac{1}{4}} \leq \frac{4}{3} T^{\frac{3}{4}}$, we get

$$
\mathbf{E}\left[\sum_{t=1}^{T}\left(f_t(\boldsymbol{x}_t) - f_t(\boldsymbol{x})\right)\right] \leq 110.6 \cdot RL\sqrt{Td^{1+\frac{2}{q\wedge 2}-\frac{2}{p}}} + 5.9 \cdot \sqrt{R}\,(\sigma+L)\,T^{\frac{3}{4}}\sqrt{\mathrm{b}_q(d)}d^{\frac{1}{2}-\frac{1}{2p}} \ . \square
$$

## D  Definition of $\ell_2$-randomized estimator

In this section we recall the algorithm of Shamir [33]. Let $\boldsymbol{\zeta}^{\circ} \in \mathbb{R}^d$ be distributed uniformly on $\partial B_2^d$. Instead of the gradient estimator that we introduce in Algorithm 1, at a each step $t \geq 1$, Shamir [33] uses

$$
\boldsymbol{g}_t^{\circ} \triangleq \frac{d}{2h}(y_t' - y_t'')\boldsymbol{\zeta}_t^{\circ} \ ,
$$

where $y_t' = f_t(\boldsymbol{x}_t + h_t\boldsymbol{\zeta}^{\circ})$, $y_t'' = f_t(\boldsymbol{x}_t - h_t\boldsymbol{\zeta}_t^{\circ})$, and $\boldsymbol{\zeta}_t^{\circ}$'s are independent random variables with the same distribution as $\boldsymbol{\zeta}^{\circ}$.