# OpenReview forum: "A gradient estimator via L1-randomization for online zero-order optimization with two point feedback"
_NeurIPS.cc/2022/Conference — NeurIPS 2022 Accept_

### Official Review · Reviewer_ewZu · 2022-07-06

**Rating:** 7
**Confidence:** 3
**Soundness:** 3 good
**Presentation:** 3 good
**Contribution:** 3 good

**Summary:**

This paper propose a new method to estimate the gradient via a zeroth-order oracle, which is especially effective in the L1 geometry. With this new estimator, this paper studies the the online zeroth-oder convex optimization under both canceling noise and adversarial noise, and also study the adaptive version of the algorithm under this setting.

**Questions:**

This work reminds me the paper "SIGNSGD VIA ZEROTH-ORDER ORACLE" https://openreview.net/pdf?id=BJe-DsC5Fm, which also use a sign opeartion for the sampled direction. But, of course, their convergence result is clearly worse than the one proposed by this work, because sign operation is not suitable for L2 gradient estimator.

**Ethics Review Area:**

["I don’t know"]

**Limitations:**

It seems to me that there is no clear limitation in this work.

**Strengths And Weaknesses:**

Strength:
1. The main contribution of this paper is to propose a new zeroth oracle gradient estimation method in the L1 geometry, which enjoys a few good properties like (1) it is theoretically better than the commonly used L2 gradient estimator expecially in the mirror descent on a simplex, which is a classical problem with L1 geometry. (2) It is computationally friendly since it only need to store a binary valued vector. From my viewpoint, this paper is very novel, and it is an easy accept for me.

---

> ### Author Response · Authors · 2022-08-02
> **Response to Reviewer ewZu**
>
> We thank the reviewer for the time and effort spent on our submission and for the positive evaluation of our work.
>
> Thank you for the reference to ZO-SignSGD algorithm. We’ll include it in the revision. That algorithm, at each step, takes a sign of the gradient estimator while it is not the case for our algorithm. We use a gradient estimator directly (with no sign transformation applied to it) and multiply it by the sign of the randomizing variable. So, both methods are quite different. Concerning the results, the authors in ZO-SignSGD paper assume Lipschitz continuity of the *gradient* but do not assume convexity, while we assume convexity and Lipschitz continuity of each $f_t$, and not of its gradient. So, a direct comparison of the results is not possible. Yet, in terms of memory efficiency our estimator is comparable with ZO-SignSGD.

---

### Official Review · Reviewer_Kfmr · 2022-07-08

**Rating:** 6
**Confidence:** 4
**Soundness:** 4 excellent
**Presentation:** 4 excellent
**Contribution:** 2 fair

**Summary:**

This paper studies online optimization problem with zero-order and two-point noisy feedback. The proposed algorithm is based on a randomized gradient estimator with the perturbation for the queried points drawn from an ell-1 sphere, followed by performing online mirror descent update. The authors show an improvement of polylog factors in the regret bound in certain cases compared to previous work, and the assumption can handle Lipschitzness with respect to arbitrary norms. An adaptive version where the parameter choice can be made independent of unknown noise variance, Lipschitz constant is also analyzed and the resulting regret is comparable to the nonadaptive version for the cancelling noise while degrade in the case of adversarial noisy feedback. Numerical experiments are also presented comparing with ell-2 randomization.

**Questions:**

- I think there's also the complexity of drawing samples from Gaussian vs. Laplace that need to be taken into consideration perhaps, given that one has smooth/strongly convex potential and the other doesn't.
- I'm not totally synced with this part of the literature, but in the case when q \neq {1,2} and under the cancelling noise assumption, are there no known algorithmic rates for this setup (I sense there should be)?


===== post author response =====

I thank the authors for addressing my comments and would like to keep my score unchanged.

**Limitations:**

Yes.

**Strengths And Weaknesses:**

I believe the result is of interest to the community and the writing is also very clear and well-structured. Most of the technical work is spent on analyzing statistical properties of the newly proposed gradient estimator, which is the main contribution of the paper. It is, perhaps, not a surprising result from an optimization algorithm point of view, but this slight improvement in the regret guarantee and generalization of various degree is a nice addition to the literature.

---

> ### Author Response · Authors · 2022-08-02
> **Response to Reviewer Kfmr**
>
> We thank the reviewer for the time and effort spent on our submission. Below we address the raised questions.
>
> 1) *Gaussian vs Laplace sampling.*  Note that for our estimator we need to sample $d$ i.i.d scaled Laplace random variables. The quantile function of each random variable is available in an explicit form (which is not the case for the Gaussian). Hence, we can simply use inverse sampling, i.e., generate $d$ uniformly distributed on $[0, 1]$ random variables and apply the quantile function. In other words, one does not need to consider any sophisticated sampling scheme due to the explicit expression of the quantile function. We shall insert this discussion in the appendix.
>
> 2) *$q \neq \{1, 2\}$.* It is indeed the case that previous works do not derive results for $q \neq \{1, 2\}$. However, we believe that with an extra work one can obtain such results for the previously proposed estimators. For example, using Theorem 1 from (Shamir, 2017) and computing the fourth moments for $\ell_q$-norm of a random variable distributed uniformly on the Euclidean sphere, one can get some convergence guarantees. We did not perform such computations but will include this discussion in the revision.

---

### Official Review · Reviewer_Evju · 2022-07-12

**Rating:** 7
**Confidence:** 3
**Soundness:** 3 good
**Presentation:** 4 excellent
**Contribution:** 3 good

**Summary:**

In this paper, the authors propose a gradient estimator (together with an analysis of online mirror descent with it) in the online optimization case where the algorithm gets to query--maybe with noise--the function at two distinct points. As in previous work, they build this estimator based on the gradients of the functions "smoothed by randomness". Their key contribution, however, is to smooth the function using noise distributed in the $\ell_1$-ball. In order to analyze the performance of this estimators they show how far the smoothed function may be from the actual function, and show a inequality (in the style of Poincaré's inequality) to bound the variance of their estimator. Finally, they show regret bounds both in cases with "simple" and "adversarial" noise, showing that it either matches or improves on previous work.

**Questions:**

- Why do you need to have $\Theta$ bounded?
- Algorithm 1 seems to be a dual averaging (or "lazy" mirror descent) algorithm. But in this type of algorithms usually we see the same step-size being applied to all the gradients at each step (that is, we would have $z_t = - \eta_t \sum_{s < t } g_s $ instead of $z_t = - \sum_{s < t } \eta_s g_s$ like what you do). I was curious if you've considered using the latter, which usually behaves better (more specifically, behaves better with the $\max_x V(x) - \min_y V(y)$ term) when you have adaptive step-sizes.

**Limitations:**

My guess is that the main technical impediment of this work is the dependency on a bounded set. The authors were also somewhat upfront about the $O(T^{3/4})$ and has a discussion whether this should be the right upper-bound or not, which is great.

**Strengths And Weaknesses:**

### Strengths
- The result is elegant and flexible, yielding already improved regret bounds and allowing for the analysis of regret bounds with quite loose assumptions on the noise;
- The Poincaré-style inequality can be a tool of independent interest for the community (as the authors mention);
- The regret analysis are clean enough for all the constants to be clearly written in the bound, which apparently was not the case with previous bounds;
- The presentation of the technical part is good (focusing on the key technical innovations) and seems sound to me.
### Weaknesses
- I know that part of this is due to space limitations, but I felt the previous/related work was poorly discussed. I am not very acquainted with the latest developments on zeroth-order (online?) optimization. Just based on the paper itself I could not get a satisfying picture to properly place the paper's contribution in the literature. I had to read the introductions/related work sections of [13] and [31] to get a better picture, which is not ideal. It makes me unsure as well, for example, if really no other work has studied adaptive optimization algorithms for zeroth-order optimization.
- I was (and still am) puzzled that the authors need $\Theta$ to be bounded, and they do not discuss why they need this in the paper. This is a clear weakness and, just from the main body of the paper, does not seem to be due the gradient estimator they use.
- (Minor) The use of references is also not good, and is related to the lack of a good discussion of related work. A list of 10 papers without any context does not help readers that are not aware of the latest work in the topic and can annoy the authors of the papers you cite.


### Summary
I think the paper is very solid and clearly deserves to be accepted. I'm not extremely familiar with the landscape in 0th-order optimization, but this feels to be potentially very impactful. Depending on the rebuttal phase and the discussion with authors and other reviewers I might increase my score.

---

> ### Author Response · Authors · 2022-08-02
> **Response to Reviewer Evju**
>
> We thank the reviewer for the time and effort spent on our submission and for the positive evaluation of our work. The reviewer mentions some weaknesses and raises some questions which we address below.
>
>
> 1) *Literature review.* Indeed, due to the space constraint, the provided exposition on related works might seem incomplete. We however note that the topic of (online) zero-order optimization is very diverse with contributions working under very different sets of assumptions. Hence, we have mentioned several papers that treat zero-order optimization and provided an expanded discussion on the state-of-the-art results obtained under assumptions closest to ours. In the revision, we shall commit to improve our presentation by giving more details on the other prior work that we cite.
>
> 2) ``[...] adaptive optimization algorithms for zeroth-order optimization.’’ We were not careful enough with the phrasing of our claim, which can cause a potential confusion. Indeed, there are some very recent works that analyse adaptive algorithms for zero-order optimization, but none of them is in our setting. In the revision we shall stress this point, providing references to other works on adaptive algorithms and highlighting the differences with our setup.
>
> 3) *Boundness of $\Theta$*:
> Note that we are mainly interested in the regret against the *best* response in hindsight, hence this assumption is important to obtain a uniform upper bound for $R$ (as it is defined in our work). Indeed, if $\Theta$ is unbounded and $V(\cdot)$ is strongly convex (w.r.t. any norm) on $\Theta$, then $V(\cdot)$ is coercive on $\Theta$ implying that $R$ is unbounded. Some previous works do not explicitly assume this condition, but they all assume that $R$ is finite, which implies that $\Theta$ is bounded (see e.g., Item 1 of Theorem 1 in (Shamir 2017) that was the state-of-the-art result in the non-adaptive case).
> Considering the setting with convex Lipschitz objectives and bounded $\Theta$ is common in the literature on zero-order online learning starting from (Zinkevich, 2003), (Flaxman, Kalai, and McMahan, 2004) and (Agarwal, Dekel, and Xiao, 2010).​
> For the current submission, our main goal was to highlight the advantageous properties of the newly proposed gradient estimator, which already improves upon the previous best results in the case of bounded $\Theta$. However, if one is interested in regret bounds against a fixed action $x$, the boundedness assumption can be relaxed for the cancelling noise setup (partially relying on the analysis given in [1]). Yet, under the adversarial noise assumption, the analysis is more challenging and we plan to investigate it using our gradient estimator in an extended version of the paper.
>
> 4) “Algorithm 1 seems to be a dual averaging (or "lazy" mirror descent) algorithm.” In fact, we consider the standard (“non-lazy”) mirror descent as can be seen from the proofs. We are sorry for the typo in the definition of our algorithm that erroneously remains copied from the earlier draft where we have used the “lazy” version. We’ll correct this typo in the revision. Thank you very much for pointing this out.
>
> [1] A. Cutkosky, Artificial constraints and hints for unbounded online learning, COLT 2019

---

> > ### Comment · Reviewer_Evju · 2022-08-05
> > **Thanks for the replies and no further questions**
> >
> > I'd like to thank the authors for the great reply to my review. And I am glad one of my confusions turned out to be on an important typo in the paper.
> >
> > As always, you should take reviews with a grain of salt. From the reply, it feels like point 2 should be prioritized and that point 1, while good for people not quite familiar with 0th-order optimization, can only be improved to a certain point without writing a complete survey on such a broad topic. So use your best judgement on what to actually write in the final version of the paper.
> >
> > On 3, what I had in mind is exactly what was mentioned in the end: whether you could relax the boundedness assumption by comparing to fixed points in hind-sight instead of only against the best (and having some dependency on the distance to the initial point in the regret bounds instead of a "diameter"). You last comment, specifically when you said "Yet, under the adversarial noise assumption, the analysis is more challenging and we plan to investigate it using our gradient estimator in an extended version of the paper" sounds super interesting and, if space allows, discussion on why the analysis gets more complicated would be much appreciated.

---

> > > ### Author Response · Authors · 2022-08-09
> > > **Reply to Reviewer Evju**
> > >
> > > We would like to thank the reviewer for their reply and valuable feedback. We will take into account your suggestions and provide a discussion on point 3.

---

### Meta-Review · Area_Chair_wQtC · 2022-08-25

**Recommendation:** Accept
**Confidence:** Certain

**Metareview:**

Overall all reviewers were positive about this paper and the overall impression is very good - accept.

**Award:**

No

---

### Decision · Program_Chairs · 2022-09-14

Accept